# Does the Choice of Anaesthesia Affect Cancer? A Molecular Crosstalk between Theory and Practice

**DOI:** 10.3390/cancers15010209

**Published:** 2022-12-29

**Authors:** Wiebrecht Debel, Ali Ramadhan, Caroline Vanpeteghem, Ramses G. Forsyth

**Affiliations:** 1Department of Anesthesiology, University Hospital Ghent, 9000 Ghent, Belgium; wiebrecht.debel@ugent.be (W.D.); caroline.vanpeteghem@ugent.be (C.V.); 2Department of Pathological Anatomy, Universitair Ziekenhuis Brussel, 1090 Brussels, Belgium; ali.ramadhan@uzbrussel.be; 3Laboratorium for Experimental Pathology (EXPA), Vrije Universiteit Brussel, 1090 Brussels, Belgium

**Keywords:** anaesthesia, carcinogenesis, cell lines, outcome

## Abstract

**Simple Summary:**

In recent years, there has been an increasing scientific interest in the interaction between anaesthesia and cancer development. Retrospective studies show that the choice of anaesthetics perioperatively may influence cancer outcome and cancer recurrence; however, these studies show contradictory results. Reviewing the recent and relevant literature for the biological effects of anaesthetics on cancer cells in comparison to the clinical effects, it was found that sevoflurane, propofol, opioids and lidocaine are likely to display direct biological effects on cancer cells. However, significant effects are only found in studies with exposure to high concentrations of anaesthetics for longer than practical durations, therefore incomparable to their clinical use.

**Abstract:**

In recent years, there has been an increasing scientific interest in the interaction between anaesthesia and cancer development. Retrospective studies show that the choice of anaesthetics may influence cancer outcome and cancer recurrence; however, these studies show contradictory results. Recently, some large randomized clinical trials have been completed, yet they show no significant effect of anaesthetics on cancer outcomes. In this scoping review, we compiled a body of in vivo and in vitro studies with the goal of evaluating the biological effects of anaesthetics on cancer cells in comparison to clinical effects as described in recent studies. It was found that sevoflurane, propofol, opioids and lidocaine are likely to display direct biological effects on cancer cells; however, significant effects are only found in studies with exposure to high concentrations of anaesthetics and/or during longer exposure times. When compared to clinical data, these differences in exposure and dose–effect relation, as well as tissue selectivity, population selection and unclear anaesthetic dosing protocols might explain the lack of outcome.

## 1. Introduction

A precise understanding of cancer remains one of the most challenging puzzles of the 21st century. Although surgery for the removal of (large) primary tumours has been regarded as one of the cornerstones in cancer therapy, the exact clinical influence of anaesthetics on cancer biology remains largely unknown. More specifically, interest in this particular subject was already raised two decades ago, and a definitive answer about its exact role in cancer has not been given yet. Several retrospective studies posed a relation between the type of anaesthetics used and the disease-free survival or cancer recurrence. In this point of view, many study reports are available in the literature, but their results are indefinite and oftentimes contradictory. For example, in vitro studies and animal experiments show some benefits for certain types of anaesthetics but solid significant results in clinical trials have yet to be supplied [1]. There are several smaller clinical trials exploring the perioperative effects of anaesthetics on cancer outcomes but these are mostly pilot studies consisting of small sample sizes. Moreover, there are major differences between these study protocols. Interestingly, three recent randomised clinical trials compared different anaesthetic protocols in which certain types of anaesthetics (neuraxial versus opioids, sevoflurane versus propofol) were studied. Nevertheless, these studies still keep lacking significant results [2,3,4]. In this perspective, the methodological aspects of all of these study designs may be possible culprits for the lack of significance. Otherwise unknown confounders might be present, as is the case when few comparable clinical trials are run in the same study field. On the other hand, it is biologically well known that the development of cancer is a multistep process. In this multistep of cancer development certain abilities must be acquired. Moreover, some of these must be kept acquired in order to survive and giving rise to metastasatic disease. Historically, six hallmarks have been described, namely resisting cell death, sustaining proliferative signalling, evading growth suppressors, activating invasion and metastasis, enabling replicative immortality, and inducing angiogenesis. More recently, other hallmarks of cancer have been added as playing a vital role in tumour progression: avoiding immune destruction, tumour-promoting inflammation, genome instability and mutation, and deregulating cellular energetics. Overlooking the recent literature, it is believed that not all hallmarks are affected by anaesthetics. There still is some evidence that certain anaesthetics could influence certain hallmarks. Next to this, it is known that immune responses are regulated by the hypothalamic-pituitary-adrenal (HPA) axis and the sympathetic nervous system (SNS). Therefore, it is of no surprise that activation of these systems induced by surgery or anaesthetics may facilitate tumour activation or distant metastatic disease as such. As for direct effects of anaesthesia, a multitude of mechanisms have been described in the literature. To our knowledge, the mapping of effects of anaesthetics on cancer development is still very limited, and oftentimes molecular pathways are incomplete. The resulting effects can be studied more easily, and the causality can be proven by in vitro knockout models of specific receptors and signalling molecules. The major aim of this review is to get a better sight on whether the currently used anaesthetics have a true direct biological and above all a significant clinical effect, or not, on the outcome during cancer surgery. Furthermore, we aim to discern confounders and methodological errors that help explain the lack of significant results. Therefore, we have opted to use the hallmarks of cancer to facilitate interpretation of the described pathways and mechanisms, and also for categorization of the resulting effects of anaesthesia on cancer cells [5].

## 2. Materials and Methods

Only publications describing the direct biological effects of anaesthetics on a cellular level, including receptors, pathways and possible mechanisms of action, were included. In addition to this, publications reporting the pathophysiological mechanisms affiliated with tumour cell growth together with a description of the causal effect of the anaesthetics were included. As the first eligible criteria, it was defined that these studies must be clinical trials or observational studies, that the in vivo or in vitro studies must be executed on human cell lines (including the use of xenografts), and that these publications describe the cellular changes during and/or after exposure to anaesthetics. Along with this, the anaesthetics discussed must be relevant and applied in current daily practise. Moreover, inclusion was limited to publication dates not before the year 2000 and to those written in the English language. In the data processed, the focus was foremost on the described biological effect of the anaesthetic agent on the type of cancer cell, including mechanism of action and the signalling pathways if sufficiently investigated. Furthermore, a subanalysis of dosage and exposure of anaesthetics was performed in comparison to the typical clinical use. Data were classified according to the type of anaesthetic agent, the type of tumour and the pathophysiological pathway of cancer initiation and metastasis. The huge amount of publications available in the literature concerning this subject made traditional relevant searches through bibliographic databases very complicated. From this point of view, relevant publications were selected based on title, abstract and full article.

These lists were filtered for double references. One reviewer was used for selection and data extraction. No tools for risk of bias were used. For supplemental data specifically on the cellular effects of propofol, sevoflurane, ketamine, lidocaine, midazolam and dexmedetomidine the bibliographical database PubMed.ncbi was used with the following search strategy: (((**[MeSH Major Topic]) AND (cancer[MeSH Major Topic])) AND (in vitro)) OR (((**[MeSH Major Topic]) AND (cancer[MeSH Major Topic])) AND (in vivo)) **: Independent search queries were made for propofol, sevoflurane, desflurane, ketamine, lidocaine, midazolam and dexmedetomidine.

## 3. Results

In short, anaesthesia and its direct effects on cancer are subdivided according to the type of anaesthetics (see Table 1).

### 3.1. The Direct Effects of Midazolam, Dexmedetomidine and Ketamine Are Insufficiently Proven

The results of the database search concerning the direct effects of midazolam, ketamine and dexmedetomidine produced the list of studies available in Table 1.

Concerning midazolam, only three studies describing its direct effects. It was stated that midazolam showed a proapoptotic effect and halts cell cycle progression. Therefore, midazolam reduces tumour growth [6,7,8]. Of interest in this perspective is that Wang et al. [6] (see Table 1) described that lower doses might not have the same effects, and since the doses used far exceed clinical use, these results cannot be extrapolated to human research. Further limitations of these studies are prolonged exposure to the drug compared to clinical use and the use of different cancer tissues, making a comparison between these studies extremely difficult.

The three studies describing the effects of dexmedetomidine show similar concerning issues. Dexmedetomidine showed an effect on apoptosis, although depending on which study was selected a pro- or anti-apoptotic effect was found. This effect is mediated through alpha 2-adrenoreceptors [6,9]. In line with midazolam, the dosage used in these particular studies is supraphysiological and the exposure time is much longer than typically is the case in an operative setting (48 h). Thus again this is limiting extrapolation to human research. Next to this, other effects are described such as a decrease in the overall survival, and in vivo in mice, and a dose-related worsening in outcome [11,12]. 

Only one study could be found that describes the direct effects of ketamine. It demonstrated a decreased apoptosis, thus promoting tumour growth. Although in vivo studies have confirmed these effects, both in vivo and in vitro very high doses of ketamine were administered [10,13]. Only one retrospective study of breast cancer describes the possible effects in a true clinical setting and this study showed no significant difference in outcome or recurrence [14].

Taking together, anaesthetics such as midazolam, dexmedetomidine and ketamine are insufficiently investigated to proclaim any possible effect on use during anaesthesia for cancer surgery.

### 3.2. The Direct Effects of Volatile Anaesthetics Are Dependent on the Type of Anaesthetic and the Biological Type of Tissue

Table 2 shows the collected data on volatile anaesthetics such as isoflurane, desflurane and sevoflurane. Five publications describing the direct effects of isoflurane on cancer cells were found. Generally, it is agreed that HIF-1A is involved which modulates the expression of VEGF-A, and angiopoietin-1, thus increasing tumour angiogenesis, glycolysis and cellproliferation [15,16,17]. Next to this, a decrease in apoptosis is described in colon tumour cells, conceivably due to the effect on caveolins [18]. These differences in direct effects may be explained by the selection of different tissues used for research as exposure and concentrations were largely similar. Very little information is available on the use of isoflurane and outcome in the true clinical setting.

Concerning desflurane only two studies were included, showing a prometastatic effect and an effect on cell cycle progression. There is also a dose dependent effect on apoptosis. More specifically, low doses induced a proapoptotic effect, where high doses in contrast demonstrated an antiapoptotic effect. Therefore, more studies are required to determine more strongly the direct effects of desflurane on cancer cells [17,19].

The effects of sevoflurane are more extensively studied resulting in an inclusion of 18 studies describing the direct effects on cancer cells. These results show varying, sometimes very contradictory results. As mentioned earlier, this is possibly due to differences in tissue selection. In lung tissue, a small reduction in viability is found. In gastric cells, sevoflurane weakens proliferative and migratory abilities through yet unknown mechanisms [20,21]. However, pro-oncogenic traits were found in renal cells, cervix cells, and head and neck squamous cell carcinomas treated with sevoflurane [20,22]. In breast cancer tissue, a small increase in cancer cells is described. This is likely caused by a change in intracellular calcium homeostasis [23,24]. In studies investigating the influence of sevoflurane on Smad3 signalling, which regulates cell proliferation, differentiation and cell death, proliferation is increased in non-small cell lung carcinomas (NSCLC) whereas in contrast it is decreased in renal cell carcinoma, therefore, producing contrary effects. Again, this further strongly supports the hypothesis that the effect of anaesthetics is fundamentally different, especially depending on the exact type of tissue.

However, even within the same series of tissue samples differences in effects were noted. Additionally, cumulative dosing over time plays an important role as well [24]. For example, in ovarian tissue exposed to sevoflurane an enhanced metastatic potential through CXCR2 is described. Next to this, also inhibition of proliferation and migration through the P38/MAPK pathway was demonstrated [17,25]. In brain tissue cell migration and invasion were repressed in two studies, and invasion potential was increased in one [26,27]. This could clearly be explained by differences in study protocols and duration of exposure times. More specifically, retrospective studies were not able to show any clinical significance in lung tissue (NSCLC) and mixed samples [28,29,30]. In breast tissue, however, retrospective studies showed contradictory results in comparison with propofol with either worse outcomes for sevoflurane [31,32] or no significant difference at all [33]. One large retrospective study by Enlund et al. showed no significant difference between propofol and sevoflurane after correction for cofounders [28]. Again, very little is disclosed about anaesthetic dosing in these retrospective studies, and study protocols are very heterogeneous. Moreover, there are no randomised clinical trials (RCT) available that prove the superiority or inferiority of sevoflurane in the daily clinical setting.

**Table 2 cancers-15-00209-t002:** Studies describing the direct effects of isoflurane, desflurane and sevoflurane on cancer cells, the associated mechanisms of action, and their respective pathways ^1^.

Isoflurane				
Study	Type of Cancer	Effect on Cancer	Mechanism of Action	Pathway Described
Benzonana L.L. et al. [15]	Kidney	Increased proliferation Cytoskeletal rearrangement Migration of cells	Increased HIF-1a and HIF- 2a expression	PI3K/Akt/mTOR pathway
Luo X. et al. [16]	Ovary	Up-regulation of markers associated with the cell cycle, proliferation, and angiogenesis	Increased VEGF, angiopoietin-1 and MMPs expression	The IGF1/HIF signalling pathway
Iwasaki M. et al. [17]	Ovary	Enhanced metastatic potential	Significant increase in mRNA for CXCR2, VEGF-A, MMP11 and TGF-β	CXCR2 plays crucial roll in the pathway, knockdown mitigates anaesthetics effect
Kawarguchi Y. et al. [18]	Colon	Resistance against apoptosis via a Caveolin dependent mechanism No effect from isoflurane alone	Resistance against TNF-related apoptosis-inducing ligand (TRAIL)-induced apoptosis via Cav-1–dependent mechanisms	Possible mechanism: Caveolins are changed in configuration due to effect on lipid membrane of volatiles
**Desflurane**				
**Study**	**Type of** **Cancer**	**Effect on Cancer**	**Mechanism of Action**	**Pathway Described**
Iwasaki M. et al. [17]	Ovary	Enhanced metastatic potential	Significant increase in mRNA for CXCR2, VEGF-A, MMP11 and TGF-β (change TME)	CXCR2 plays a crucial role in the pathway, and knockdown mitigates anaesthetics
Bundscherer A.C. et al. [19]	Colon	Affection of cell cycle regulation Affection of apoptosis after 6 h exposure	Non described	Non described
**Sevoflurane**				
**Study**	**Type of** **Cancer**	**Effect on Cancer**	**Mechanism of Action**	**Pathway Described**
Ecimovic P. et al. [23]	Breast	Small increased proliferation and migration	Non described	Non described
Deng, X. et al. [24]	Breast	Sevoflurane, but not propofol, at clinically relevant concentrations and durations: increased survival of breast cancer cells in vitro no effect on cell proliferation, migration or TRPV1 expression.	These findings suggest that changes in intracellular Ca2+ homeostasis play an important role in the general anesthetic-mediated enhancement of breast cancer cell survival	The TRPV1 channel is a potential site of action of sevoflurane in altering intracellular Ca2+ levels
Iwasaki M. et al. [17]	Ovary	Enhanced metastatic potential.	Significant increase in mRNA for CXCR2, VEGF-A, MMP11 and TGF-β	CXCR2 plays a crucial role in the pathway, and knockdown mitigates anaesthetics
Kang K. et al. [25]	Ovary	Inhibition of cell proliferation, migration and invasion, and induced apoptosis of the OC cell line	PCNA, Twist, MMP-2 and MMP9 mRNA expressions were significantly decreased while caspase-3 expression was markedly increased in sevoflurane groups compared to that in the control group	Dramatical decrease of p-p38/p38 and p-JNK/JNK expressions in OC cells of sevoflurane groups compared to that of the control group, important in p38 MAPK Signaling Pathway
Ciechanowicz S. et al. [20]	Lung (NSCLC)	Reduced cell viability No effect on metastatic potential	Unchanged levels of TGF-b1, possible homeostatic regulation/sensitization	Upregulation of Smad3 signalling
Ciechanowicz S. et al. [20]	Renal cell carcinoma	Increases cell viability Promotes metastatic potential	TGF-b1 plays a role in cytoprotection, proliferation and migration	TGF-b and OPN upregulation. Reduced nuclear Smad3
Ferrell J.K. et al. [22]	Head and neck SCC	Increase in the expression of pro-oncogenic protein markers	Exact mechanism unclear	Statistically significant increases in the expression of cytoplasmic HIF-2a and nuclear p-p38 MAPK
Bundscherer A.C. et al. [19]	Colon	Affected cell cycle regulation Increased apoptosis in low and high doses	Non described	Non described
Chen H. et al. [21]	Gastric	Weakening proliferative and migratory abilities	Exact mechanism unclear	Upregulation of miR-34a/TGIF2 axis
Zhao H. et al. [26]	Brain	Repressed cell migration and invasion	Upregulation of miR-34a-5p, which inhibits MMP-2 thus reducing metastasis	Non described
Lai R.C. et al. [27]	Brain (glioblastoma)	Exposure to 1–4% sevoflurane did not change the cell proliferation Concentration-dependent increasement of invasion of human glioblastoma U251 cells	Increased activity of calpains, a group of cysteine proteinases, and CD44 protein	CD 44 regulates intracellular signalling, unsure which pathway is involved

^1^ The correlated hallmark of cancer is given for reference on the role and importance in cancer development. Sorted by type of tissue.

### 3.3. The Direct Effects of Propofol Show a Multitude of Cellular Changes Dependent on Tissue Type

This study has yielded several studies describing the direct results of propofol on cancer cells. When comparing the direct effects per type of tissue, the most frequent described effects of propofol are anti-oncogenic and anti-metastatic (see Table 3). An exception is a study on prostate cancer, where no effect was found [18]. The mechanisms linked to these effects varied greatly, again mostly and especially depending on the type of tissue used. The most important effects are inhibition of matrix metalloproteinases (MMPs) causing inhibition of invasion, EMT, and metastasis [34,35,36,37] and increased apoptosis [33,38,39,40,41]. Other studies reported an effect on the cell cycle [39,40,41,42].

Overall, it is clear that propofol directly affects cancer cells through a multitude of translational changes and modulation of cellular pathways. However, there are a considerable amount of important limits to these studies. First, most studies describe long periods of exposure to propofol (more than 24 h) before any significant outcomes can be found. Furthermore, the study protocols implemented exposure to a relatively high concentration of propofol. One study reported some significant effects started clinically at a relevant dose up to 5 µg/mL [43], but most significant effects started at 10 µg/mL or even higher. In shorter periods of exposure or at lower concentrations, these studies show no significant effect at all. This is of particular importance to the reference of the clinical use of propofol, in which typically shorter periods of exposure are custom during cancer surgery. Although exposure periods of 24 h may be clinically applied, concentrations of 5 µg/mL are rarely used during sedation in the intensive care unit. Currently, there are no RCTs available that could show anti-oncogenic effects of propofol with differences in the outcome on overall survival or disease-free survival in the clinical setting. Next to this, there is a multitude of retrospective studies with varying outcomes. Most describe any significant difference in outcome for propofol versus other anaesthetics in a collection of tissue samples [33,44,45,46]. Others described superior effects of propofol versus volatile anaesthetics in the breast [31,32] hepatocellular [47] or mixed [33,48] tissue samples. As was mentioned earlier, a large difference in anaesthetics and study protocols is clearly noted in comparison of these studies as well as the presence of multiple confounders and biases that inherently trouble retrospective research seriously. More specifically, the concentration of anaesthetic used, combinations of possible influential drugs used perioperatively and the time of exposure may be important factors. Very little about this is documented in these retrospective studies.

**Table 3 cancers-15-00209-t003:** Studies describing the direct effects of propofol on cancer cells, the associated mechanisms of action and their respective pathways ^1^.

Propofol				
Study	Type of Cancer	Effect on Cancer	Mechanism of Action	Pathway Described
Yang C. et al. [39]	Gastric	Significantly inhibited cell proliferation, invasion and migration Enhanced apoptosis	Upregulation of expression of inhibitor of growth 3 (ING3)	Non described
Gao J. et al. [42]	Lung (NSCLC)	Suppression of cell proliferation, invasion and glycolysis Expedited apoptosis cells	FOXM1 (transcription factor belonging to the Forkhead box family) essential for cell cycle progression	circ-ERBB2/miR-7-5p/FOXM1 axis
Ecimovic P. et al. [37]	Breast	No effect on proliferation Reduced migration and reduced invasion of MCF7 but not of MDA-MB-231 cells.	The Neuroepithelial Cell Transforming Gene 1 (NET1) gene is associated with promoting migration in adenocarcinoma in vitro.	Propofol reduced expression of NET1
Garib V. et al. [43]	Breast	Activation of GABA-A receptor correlated with an increased migration of MDA-MB-468 breast carcinoma cells	Mediated by calcium influx and reorganization of the actin cytoskeleton	GABA-A receptor activation results in activation of voltage-gated L-type calcium channels
Li Q. et al. [34]	Breast	Inhibition of the invasion and migration of breast cancer cells	Down-regulation of MMP-2 and MMP-9	Reduction phosphorylation of IKKb (Ser180), which is an important upstream kinase for IkB degradation and subsequent NF-kB activation
Su Z. et al. [40]	Ovary	Inhibition of proliferation and induction of apoptosis	Exact mechanism unclear	Increased expression of miR-let-7i, no pathway described
Lu H. et al. [41]	Ovary	Suppression of proliferation, cell cycle, migration and invasion Induction of apoptosis of ovarian cancer cells	Up-regulating miR-145 via down-regulating circVPS13C	Inhibited the activation of MEK/ERK signalling
Takabuchi S. et al. [49]	Hepatocellular	Suppression of HIF-1a protein expression was significant at 20%and 5% O_2_ but not at 1% O_2_	Suppression of the translation of HIF-1amRNA into protein	Possible role of MAPK pathway
Zhang J. et al. [38]	Hepatocellular	Induction of apoptosis of HCC cells Modulation of miR-199a contributing to the antitumor action of propofol	Induction of apoptosis and activation of caspase-8 and caspase-9 in a dose-dependent manner	Stimulation of miR-199a expression in HepG2 cells
Zhang J. et al. [36]	Hepatocellular	Inhibition of the invasiveness of HepG2 cells	Significant decrease in expression of MMPs Significant inhibition of the activity of MMP-9 in HepG2 cells but no effect on MMP-2 expression	Exact mechanism unclear, possible role of miR-199a
Zhang D. et al. [47]	Cervix	Inhibition cervical cancer cell growth Induction of apoptosis	Decreased HOTAIR expression of cervical cancer cells in a dose-dependent manner. HOTAIR is an lncRNA which is noticeably increased in multiple carcinomas.	HOTAIR activates mTOR/p70S6K pathway leading to cell growth
Zhang L. et al. [50]	Glioma	Repression of cell growth and metastasis in glioma cells in vitro and in vivo	Exact mechanism unclear	Mediated by the circNCAPG/miR-200a-3p/RAB5A axis
Xu Y.B. et al. [35]	Esophagus	Significant promotion of cell apoptosis and inhibition of proliferation, invasion and angiogenesis in a dose and time-dependent manner	Down regulation gene expression and protein production of VEGF and MMP-9	ERK/VEGF and ERK/MMP-9 signalling pathways

^1^ The correlated hallmark of cancer is given for reference on the role and importance in cancer development and sorted by type of tissue.

### 3.4. The Direct Effects of µ-Receptor Opioids on Cancer Cells Remain Uncertain and Are Difficult to Explore

Sixteen articles reporting the direct effects of opioids on cancer cells could be selected by this study (see Table 4). Most frequently described are the effects on breast, lung or colon tissue. However, these results are often contradictory, and multiple mechanisms varying from apoptosis, EMT and increased proliferation or activation of microenvironment and inflammation are found as direct effects. For example, some authors claimed no effect of morphine in breast tissue [51,52,53], where others described possible anti-oncogenic effects [54,55,56,57,58]. In lung tissue, the same conflicting results between oncogenic [59,60] and anti-oncogenic [56,61] effects were found. Little outcome was found in colon tissue inclusive one article describing an anti-invasive effect [55,62,63]. Large variations in outcome were noticeable for these in vitro trials. The trials describing apoptosis and necrosis did use high supraphysiological doses of morphine, possibly partially explaining the difference in outcome [56,63]. In addition, there are large differences when comparing cell lines of similar tissue. Some authors used commercially available cell lines, while others used samples of cells in their own microenvironment. Even within the same cell line set-up conflicting evidence was found [59,61].

There are no RCTs demonstrating an effect of µ-receptor opioids on overall survival or disease-free outcome. Some RCTs and experimental studies described an immunomodulatory effect of opioids and a possible decrease in natural killer (NK) Cell function and neutrophils [4,64,65]. However, a weakness in these studies is the lack of knowledge of influence of opioids and immune function together with the surgery related stress response. As both affect immune function, and stress levels are hard to objectify, this may be an extra challenge in interpreting these results as reliable.

Two retrospective studies described a correlation between the amount of µ-opioid receptors (MOR) and clinical outcomes [66,67]. However, in the case of NSCLC, retrospective studies could not show significant differences in the risk of cancer recurrence. Only a small decrease in overall survival for stage I NSCLC patients could be noticed [68,69,70]. Moreover, no impact could be demonstrated on the overall survival and disease-free survival in colorectal cancer [69]. An important possible cofounder in these cases is the chronic use of opioids during cancer treatment and the impact on the outcome.

Generally, it is largely uncertain if opioids directly affect cancer cells in vivo. Stronger paracrine and endocrine effects play a vital role in this type of research with opioids. Factors such as the effect of the tumour microenvironment, inflammation and stress response make it extremely difficult to interpret study results adequately and reliable.

**Table 4 cancers-15-00209-t004:** Studies describing the direct effects of MOR opioids on cancer cells, the associated mechanisms of action and their respective pathways ^1^.

µ-Opioid Receptor Agonists				
**Study**	**Type of Cancer**	**Effect on Cancer**	**Mechanism of Action**	**Pathway Described**
Nguyen J. et al. [51]	Breast	No influence on initiation Increased progression Decreased survival	µ-opioid receptors on large tumours, possibly stimulated by VEGF and cytokines	Stimulation of MAST cells in tumours, release of substance P, increased pain and inflammation
Doornebal C. et al. [52]	Breast	Analgesic doses of morphine do not affect mammary tumour growth	Difference in effect is explained by heterogeneity of primary tumour compared to selected cell lines chosen for metastatic potential	Non described
Afsharimani B. et al. [54]	Breast	Reduced expression of matrix-degrading enzymes in cocultures with macrophages or endothelial cells	Reduced the level of MMP-9 and increased its endogenous inhibitor, TIMP-1	Modulation of paracrine communication between cancer cells and non-malignant cells in the tumour microenvironment
Gupta K. et al. [57]	Breast	Statistically significant increase in tumour volume and vascularization Significantly increased migration in vitro	Increased vascularisation similar to VEGF Increased growth through inhibition of apoptosis	Stimulation of the MAPK/ERK signalling pathway and activation of the cell survival signal Akt and increasing cell cycle protein cyclin D1
Tegeder I. et al. [55]	Breast	Morphine significantly reduced the growth of MCF-7 and MDA-MB231 tumours	Inhibition of cell cycle progression in low dose, activation of apoptosis in high doses	p53 activation and up-regulation of p53-dependent genes (including CD95/Fas)
Ecimovic P. et al. [58]	Breast	Increase in both expression of NET1 and cell migration but not when NET1 was silenced	The NET1 gene has a key role in organization of the actin cytoskeleton and thus in the ability of cancer cells to migrate and invade	Mechanism unclear
Gach K. et al. [53]	Breast	Increase in secretion of urokinase plasminogen activator, no results on migration	Opioid agonists greatly increase the secretion of uPA from MCF-7 human breast cancer cells, as well asup-regulate the expression of uPA and uPAR genes	Through MOR increases the expression of uPA and uPAR
Hatsukari I. et al. [56]	lung (NSCLC) and breast	A clinical concentration of morphine induced apoptosis and necrosis in human tumour cell lines	Through activation of opioid receptors, no clear arguments for different mechanisms (control group with naloxone)	Non described
Mathew B. et al. [59]	Lung (NSCLC)	Increased MOR expression in NSCLC Morphine increases cell growth Methylnaltrexone decreases cell growth and invasion	Knockout of MOR receptor reduces cell growth and metastasis	Direct inhibition of MOR and activation of tyrosine phosphatase activity
Koodie L. et al. [61]	Lung/ Ovaries	Inhibition of migration of tumour infiltrating leukocytes Decrease in angiogenesis	Altering cell adhesion molecule expression on both the leukocyte and endothelial cells. Impairs mobilization of endothelial progenitors and neutrophils, thus decreasing inflammation and angiogenesis	The mechanism is unclear Possible mechanism: decreases the tight junction protein zonula occludens protein 1 expression.
Lennon F.E. et al. [60]	lung (NSCLC)	The data suggest a possible direct effect of MOR on opioid and growth factor-signalling and consequent proliferation, migration and epithelial mesenchymal transition (EMT) during lung cancer progression. DAMGO, morphine and fentanyl were used as MOR agonists	MOR regulates opioid and growth factor-induced EGF receptor signalling through Grb2-associated-binding protein 1 (Gab-1)	Activation of Src, Gab-1, PI3K, Akt and STAT3
Tegeder I. et al. [55]	Colon	No significant effect on HT-29 tumour growth	Non described	Less expression of P53, therefore less effect
Nylund. et al. [62]	Colon	Morphine largely fails to affect the proliferation of the HT-29 cell line, but causes a markedly increased secretion of uPa. No results on migration	uPa plays an important role in activating invasion and metastasis	Through MOR
Harimaya Y. et al. [63]	Colon	Significant reduction of the number of tumour colonies and of the weight of the tumour-containing lung Inhibition of adhesion and migration of cells to the extracellular matrix, without affecting the cell proliferation in vitro	Suppression of tumour cell adhesion, invasion and migration, partly through opioid receptors, partly through reducing enzymatic degradation of the ECM	Inhibition of the production of MMP-2and MMP-9 in tumour cells, no clear pathway was described
Friesen C. et al. [71]	Glioblastoma	Activation of opioid receptors sensitizes glioblastoma cells for therapy	Opioid receptor signalling pathway is involved in apoptosis induction by chemotherapy.	Opioid receptor stimulation activates inhibitory Gi-proteins, which, in turn, block adenylyl cyclase activity, reducing cAMP. Downregulation of Bcl-x and XIAP

^1^ The correlated hallmark of cancer is given for reference on the role and importance in cancer development. Sorted by type of tissue.

### 3.5. Lidocaine Affects Cancer Cells Directly through Various Mechanisms

Fifteen articles were included demonstrating the direct effects of lidocaine (see Table 5). Of particular interest is that lidocaine showed the highest concentrations perioperatively among all modern local anaesthetics, thus most likely to produce a significant observable clinical effect. Other local anaesthetics are mainly used perioperatively as part of the locoregional anaesthesia procedure, thus limiting the concentration. Like propofol, various mechanisms and pathways are described as direct effects of anaesthesia. Interestingly, all these described effects are anti-oncogenic, making lidocaine a prime candidate for further investigation. The most recurring effects were the evasion of growth suppressors [72,73], increased activation of apoptosis mechanisms [74,75,76], and decreased invasion, EMT and/or metastasis [77,78,79].

Unfortunately, all these in vitro studies used very high concentrations of local anaesthetic drugs. If extrapolated to the daily clinical use, the concentrations used are approximate to that applied to local anaesthesia with infiltration approximate to the target tissue, whereas the clinically used concentrations for locoregional or intravenous anaesthesia are considerably lower. Chamaraux-Tran. et al. (see Table 5) described a difference in sensitivity for local anaesthetics in different cell lines. Additionally, differences were noted concerning toxic effects in higher concentrations, which were frequently applied in other studies [80]. The studies that had used lower concentrations closer to clinically relevant intravenous doses (lower or equal to 10 µM) reported no significant effects at all [72,74,75,78,81,82,83,84,85]. At low concentrations, long exposure times were absolutely needed before significant results could be found. Yang. et al. (see Table 5) found significant results at 10 µM concentration, but only starting after 24 h of exposure time, and also depending on the type of cell line used [73]. Lirk. et al. (see Table 5) described a significant effect starting at least 24 h of incubation time but only for specific breast cell lines. This fits again with the hypothesis of different levels of sensitivity per cell line [81]. A few studies, however, found significant effects at lower concentrations in line with the exposure time comparable to routine clinical use [76,82]. Interestingly, in three of these low concentration studies, breast tissue was involved.

No RCTs have currently proven the superiority of lidocaine over other anaesthetics or analgesics. Zhang. et al. published a large sample retrospective study that claims a potential anti-oncogenic effect of lidocaine in pancreatic cancer with increased overall survival [86]. In vivo studies with mice, xenograft models showed a possible diminishing effect on metastasis as well [87,88].

From this point of view, it is plausible that lidocaine displays several anti-oncogenic effects by direct interaction with cancer cells. However, once again, concentrations used in these studies strongly deviates from daily clinical application.

**Table 5 cancers-15-00209-t005:** Studies describing the direct effects of Lidocaine on cancer cells, the associated mechanisms of action, and their respective pathways ^1^.

Lidocaine/Local Anaesthetics (-Ester and -Amide)				
**Study**	**Type of Cancer**	Effect on Cancer	Mechanism of Action	Pathway Described
Piegeler T. et al. [77]	Lung (NSCLC)	Both ropivacaine and lidocaine blocked tumour cell invasion and MMP-9 secretion	Attenuation of Src-dependent inflammatory signalling events	Src-dependent activation of Akt and focal adhesion kinase (FAK) and phosphorylation of caveolin-1 (Cav-1) by Src, resulting in reduced MMP-9 synthesis
Piegeler T. et al. [78]	Lung (NSCLC)	This study indicates that amide-, but not ester-linked local anaesthetics may inhibit migration of tumour cells	Independent mechanism of voltage gated Sodium channel inhibition	The inhibition of Tumour Necrosis Factor-α-induced Src-activation and Intercellular Adhesion Molecule-1 (ICAM1) phosphorylation
Chamaraux-Tran T.N. et al. [80]	Breast	Reduction in tumour viability, cell growth and migration in vitro Reduction in cell growth and increased survival in vivo	Non described	Non described
Lirk P. et al. [81]	Breast	Demethylation of DNA of breast cancer cell lines in vitro (in clinical relevant concentrations)	No effect on three known tumour suppressor genes (RASSF1A, MYOD1 and GSTP1) Demethylating effects are dependent on the type of cancer cell	Methylation of DNA changes epigenetic expression which affects expression
Li R. et al. [82]	Breast	Significant cytotoxic effect in high concentrations(1 mM), none in physiological concentrations (10 µM). arrest of MDA-MB-231 cells in the S phase for both concentrations. Most significant effect was found in the levobupivacaine group.	Non described	Non described
D’Agostino G. et al. [79]	Breast	Inhibition of CXCL12-induced in vitro migration of MDA-MB-231 cells	Lidocaine, in clinical concentrations, inhibits CXCL12-induced CXCR4 signalling, which impairs the essential cascade of cytoskeleton remodelling, leading to a reduced migration of breast cancer cells Lidocaine treatment promotes upregulation of CD44 expression (a transmembrane glycoprotein important for cancer interaction with hyaluronic acid), an essential component of the extracellular matrix	Exact mechanism unclear
Xing W. et al. [74]	Hepato-cellular	Suppression of tumour growth and induction of apoptosis in human HepG2 cells in vitro In vivo, lidocaine not only suppressed hepatocellular carcinoma development but also sensitized hepatocellular carcinoma to cisplatin	Increased ratio of Bax/Bcl-2 Increased caspase 3 activation Activation of apoptosis	Phosphorylation of ERK1/2 and P38 through the MAPK pathway
Le Gac G. et al. [83]	Hepato-cellular	Local anaesthetics decreased viability and proliferation of HuH7 cells and HepaRG progenitor cells	Ropivacaine stops the G2 phase of the cell cycle in HCC cells by decreasing key cell cycle regulators Increased apoptosis marked by increased caspase 3	Lidocaine increased mRNA levels of APC and of DKK1, which both act as antagonists of the Wnt/β-catenin pathway Ropivacaine decreased the mRNA level of cyclin A2, cyclin B1, cyclin B2, and cyclin-dependent kinase 1, and the expression of the nuclear marker of cell proliferation MKI67
Zhao L. et al. [84]	Hepato-cellular	Repression of hepatocellular carcinoma cell proliferation, migration, and invasion Promotion of apoptosis	Unclear	Via regulating circ_ITCH/miR-421/CPEB3 axis.
Liu H. et al. [85]	Hepato-cellular	Decrease in HepG2 cell viability and colony formation in a dose-dependent manner	Unclear	CPEB3 as a critical mediator of lidocaine-induced repression of HepG2 cell proliferation
Bundscherer AC. et al. [72]	Colon	Induction of cell-cycle arrest in both colon carcinoma cell lines in vitro, but no effect on apoptosis Small increase in proliferation between 10–100 µM lidocaine in SW480 cells	Cell cycle arrest, exact mechanism unclear	Unclear
Yang W. et al. [73]	Gastric	Lidocaine and ropivacaine inhibited the proliferation of AGS and HGC-27 cells within 72 h. Especially lidocaine at doses of 10 μM or above (which is safe as the blood level for clinical use)	Significant reduction of expression of p-ERK1/2 in AGS and HGC-27 cells	MAPK pathway
Ye L. et al. [75]	Gastric	Significant suppression of proliferation, migration and invasion Induction of apoptosis in a dose-dependent manner in human gastric cancer cells	Simultaneous p-p38 increasement, while the level of p38 was not affected Increased Bax level and decreased Bcl-2 level in a dose-dependent manner	MAPK pathway
Sakaguchi M. et al. [89]	Tongue	In a clinical concentration of lidocaine (400 μM): suppressed proliferation and without cytotoxicity In a larger concentration of lidocaine (4000 μM): cytotoxicity with an antiproliferative effect	Inhibition of EGF-stimulated tyrosine kinase activity of EGFR	Direct inhibition tyrosine kinase activity
Bezu L. et al. [76]	Colon, breast, cervix, osteosarcoma, fibrosarcoma	Lidocaine and other anaesthetics induced signs of cancer cell stress including inhibition of oxidative phosphorylation, and induction of autophagy as well as endoplasmic reticulum (ER) stress	Induction of ER stress, resulting in eIF2α phosphorylation, causing activation of autophagy	EIF2AK3/PERK-dependent eIF2α phosphorylation leading to ATF4 translation, IRE1-mediated XBP1 activation, as well as activation of the latent transcription factor ATF6

^1^ The correlated hallmark of cancer is given for reference on the role and importance in cancer development. Sorted by type of tissue.

## 4. Discussion

In the multistep development of human cancers, certain abilities must be acquired along the way in order to fully differentiate as a malignant cancer cell. Next to this, some of these abilities must be kept acquired in order to be able to further survive and metastasize. These abilities are at present times best described by the model known as the hallmarks of cancer, specifying the particular importance of the different mechanisms discerning malignant cells from healthy cells. Historically, six hallmarks have been described, namely resisting cell death, sustaining proliferative signalling, evading growth suppressors, activating invasion and metastasis, enabling replicative immortality, and inducing angiogenesis. More recently, other hallmarks of cancer have been suggested as playing a vital role in tumour progression. Hallmarks such as avoiding immune destruction, tumour-promoting inflammation, genome instability and mutation, and deregulating cellular energetics. Where the abilities to sustain proliferative signalling and evade growth suppressors lie at the start of the development of a malignant cell, by dividing without limitation, other hallmarks such as genome instability and mutation, resisting cell death, and enabling replicative immortality help it by further differentiating from regular cells and mutating more favourable genes. The other hallmarks help to further amplify growth signalling, regulate the delivery of nutrients and oxygen, and create a favourable microenvironment for the tumour to thrive in. This includes increasing vascular growth, recruiting stromal cells for further positive signalling, and manipulating the immune response to gain a more favourable outcome for the cancer cells. Finally, the ability to activate invasion and metastasis helps the spread to further satellites and increases the mortality of cancer itself [5]. In this point of view, cancer cells are present in a complex tissue microenvironment whereby there is an interaction of surrounding different types of cells including noncancerous cells, cells of the immune system, the extracellular matrix, chemokines, cytokines, and other factors [90]. The microenvironment provides positive growth signalling but manages also energy delivery through vascular growth and immune modulation. Interactions between circulating tumour cells and the microenvironmental components of circulation determine survival and the ability of these cells to eventually extravasate in distant sites [91,92]. A final step in differentiating between primary tumour and metastasis is the induction of epithelial to mesenchymal transition (EMT), where the cell loses its polarity and cell–cell adhesions through multiple biological transformations and changes into a mesenchymal cell type, facilitating migration and invasion, ultimately resulting in increased metastatic potential [93]. Spontaneous epithelial-mesenchymal transition EMT in primary tumour cells shifts between different intermediate stages with different characteristics [94]. It has been reported that the EMT programme is a gamut of transitional steps between the epithelial and mesenchymal phenotypes. In fact, studies suggest that the nature of the primary cancer cell determines the different metastatic properties with respect to growth and response to therapy [95,96].

Generally, not all hallmarks seem to be affected by anaesthetics, but there is evidence that some anaesthetics are able to influence certain hallmarks. As it is known that immune responses are regulated by the hypothalamic-pituitary-adrenal (HPA) axis and the sympathetic nervous system (SNS), it is no surprise that activation of these systems induced by surgery or anaesthetics may facilitate tumour activation or distant metastasis through several tumour-derived soluble factors suppressing the HPA axis, activating SNS cellular immune responses (CMI) and releasing catecholamines and prostaglandin E2 (PGE2) [97]. Subsequently, these factors will increase immunosuppressive cytokines, and soluble factors (for example interleukin-4 [IL-4], IL-10, transforming growth factor Beta [TGF-β], vascular endothelial growth factor VEGF]), and pro-inflammatory cytokines (e.g., IL-6 and IL-8), and ends up promoting angiogenesis and metastasis [98,99,100,101].

It is known, and widely accepted that a number of factors that occur in the perioperative period have a significant and direct impact on the status of cancer cells on the one hand and the body’s cellular immunity on the other hand. This concept enhances the possibility of cancer spreading or reactivating cancer cells from a state of inactivity and dormancy, and turning them into functional and active cancer cells. Since the perioperative changes to immune function, inflammation, stress response and cancer activity are complex and multifactorial, there has been significant difficulty in describing causality or associative connections between anaesthesia and cancer, especially since microenvironmental regulation and interactions between host reactions and cancer cells are difficult mechanisms to recreate in a research environment. A limitation of this study is that only the direct cellular effects of anaesthetics on tumour mechanisms and development were evaluated through extensive research of literature. Recent clinical trials have tried to study the multifactorial indirect effects of anaesthesia during surgery. Comparing perioperative epidural analgesia with opioid analgesia in thoracic and abdominal surgery showed no difference in overall survival and disease-free survival either [2,3]. The anaesthetic protocols of the latter study were matched for the hypothesis that the effect of opioids might be immunomodulatory and that locoregional anaesthesia could possibly inhibit stress response in major surgery. However, both studies allowed for opioids to be used in epidural groups, and showed similar opioid consumption. When the proposed mechanisms of those studies were compared to literature search findings of this review, several conclusions can be made. Since this study was limited to the direct effects of anaesthetics, we cannot exclude a possible effect of the shift in immune function or changes in stress response. Stress response refers to the situation in which a complex series of physiological events occurs following an injury or trauma. It is, therefore, best defined as the natural response to physiological stress that occurs as a result of surgery and associated perioperative events. Beside the activation of the neuroendocrine system, there will be some inflammatory changes and activation of the hypothalamic-pituitary-adrenal axis. The complex interplay of these events, in turn, leads to the formation of a constellation of immune, hemodynamic and metabolic changes. Surgical stress is induced by the release of norepinephrine and epinephrine that may interact with β1 and 2 receptors expressed by cancer cells increasing their invasive and proliferative capacity [98,102,103]. Although it has been believed that the stress response plays a beneficial role in recovery and survival after injury, several publications have appeared in recent years documenting that with regard to cancer recurrence and metastases after surgery, the effect of the stress response may not be beneficial. Certain elements of the stress response are thought to boost cancer growth [1]. In addition, the inflammatory component of the stress response includes the production and release of cytokines, prostaglandins and cyclooxygenase. The chronic release of such mediators are thought to play a role in cancer development through inhibition of apoptosis, promotion of angiogenesis and immunosuppression [104]. It has also been suggested that acute release of such mediators during the perioperative period may promote cancer growth [105]. Finally, pain activates the hypothalamic-pituitary-adrenal (HPA) axis, which has been implicated in immunosuppression, reduction of natural killer cell activity and enhancement of tumour cell activity in animals [106,107].

Forget. et al. showed that surgical trauma in rats works as a very powerful metastatic stimulus [13]. Since greater surgical stress is associated with more pain and worse surgical outcome, it is very difficult to objectively measure its role on cancer in human subjects. To our knowledge, there are no human trials on the causality of stress response and cancer outcome. Surrogate markers can be used such as postoperative pain, changes in vital signs attributable to sympathetic stimuli or endocrine measurements. In animal studies, there is evidence to back up this theory [12]. For pain and postoperative morphine consumption, there were no significant differences in the comparison of epidural and opioid analgesia, and no data on stress hormones, thus making it difficult to interpret the effect of the stress response on outcome. The lack of arguments for changed stress responses in all three RCTs might explain the lack of significant results [2,3,4]. Opioids may display an immunomodulatory effect, as mentioned earlier. They possibly decrease NK cell function and modulate the function and differentiation of T cells. It is well known that NK cells play a major role in the first immune defence against malignancy and that factors such as neutrophil-to-lymphocyte ratio (NLR) have a negative prognostic value in cancer outcomes. In this point of view, morphine suppresses neutrophil functions such as phagocytosis, respiratory burst, and complement receptor expression by stimulating NO release via μ3 receptors [108]. T lymphocyte functions and B lymphocyte functions are also suppressed by morphine in vivo. The mitogenic response of B-lymphocytes plasma cells [109] is suppressed by morphine administration in vivo. Moreover, T-lymphocyte proliferation is decreased by both acute and chronic morphine administration [110]. In contrast to the inhibitory effects induced by morphine on immune cells, synthetic opioids such as fentanyl and remifentanil have no effect in attenuating immune cell responses because of a reduced interaction of synthetic opioids with specific opioid receptors. Fentanyl, remifentanil, and alfentanil do not impair functions of neutrophils such as respiratory burst [111] and phagocytosis [112] Sevoflurane may also affect NK cell function and has an immunomodulatory effect [13,113,114,115,116]. However, the effect of sevoflurane on immune function is unsure, as some studies cannot find any difference in immune function [33,117]. Furthermore, local anaesthetics in high concentration have recently been proven to trigger T lymphocyte-dependent tumour growth reduction through ER stress induction and eliciting other immunostimulatory stress signals including the release of ATP and HMGB1 from cancer cells [76]. These finding, together with the abovementioned possible changes in stress response further complicate translation of in vitro studies to the in vivo setting, and interpretation of aforementioned RCT’s.

To the best of our knowledge, no clinical study has shown the direct effect of immunomodulation by opioids or sevoflurane on cancer outcomes in a surgical setting. Xu. et al. have shown that serum treated with opioids and sevoflurane has a detrimental effect on cancer development compared to serum treated with propofol and locoregional anaesthesia, but only in vitro [65]. In summary, it is unsure what exactly the importance of immunomodulation is in the perioperative setting on cancer outcomes taking the direct effects of anaesthetics into account. However, since previously mentioned RCTs have not produced significant results, it is possible that the effect of this immunomodulation is nearly as important as previously thought, especially compared to the larger effect of the surgical stress response.

When considering the direct effects of sevoflurane, it can be stated that several discrepancies exist throughout the body of evidence concerning the effect of sevoflurane on cancer cells. It is possible that sevoflurane increases angiogenesis through mechanisms such as HIF1a; however, this is uncertain. Differences between tissue selection, time of exposure or concentrations used may explain these discrepancies. The concentration of sevoflurane turns out to be especially important since Deng et al. described a duality in action depending on the concentration of sevoflurane, where higher doses were more cytotoxic [24]. Since most recent clinical studies leave sevoflurane dosage to physician’s discretion, it might be a possible cofounder skewing results.

When overviewing the results of the performed literature search, there are several arguments to suspect possible effects on cancer cells by the anaesthetics researched. Very few of these mechanisms are evaluated in current RCTs. Additionally, there is retrospective evidence that the use of some anaesthetics would be beneficial in cancer surgery. Therefore, it is very surprising that no benefits have been found in recent randomised clinical trials. A number of explanations can be formulated concerning these results. The first is the absence of effect on cancer cells in study designs with low concentrations of anaesthetics and short periods of exposure. We observed that most of the anaesthetic study drugs were used in vitro at supraphysiological concentrations with long exposure times. Although this makes sense for in vitro studies to increase the chance of significant outcomes, it makes translation to in vivo research very challenging. Most studies report lack of significant outcome when doses are lowered or exposure time is shortened such as observed with midazolam, dexmedetomidine, lidocaine and propofol [6,10,34,43,72,74,75,78,81,82,83,84,85,109]. There are studies that prove possible effects in physiological doses and exposure time for lidocaine, but to a lesser extent [79,87]. The effects of opioids in anaesthesia for cancer surgery are highly debatable, since direct effects in vitro are contradictory, and the chronic use of opioids is frequent in cancer patients. It may be possible that the immunomodulatory effects play a role in cancer outcome, but this is not clearly proven in clinical studies so far. Sevoflurane is inconclusive as well due to the duality of its effect and lack of transparency on dosing protocols in clinical studies. Thus, a possible scenario could be that when these anaesthetics are used in clinically relevant concentrations and duration, there is an absence of a true direct effect on cancer cells. The second explanation for these conflicting results found in literature is the complex interaction between tumour and microenvironment and the biochemical changes of EMT. These are complex physiological interactions that are very hard to recreate in vitro, thus making research more complicated and difficult. Additionally, it may have a huge effect on the behaviour of cancer exposed to anaesthetics. This microenvironment is easily perturbed by any tissue trauma, and surgical intervention aimed at eliminating the disease may unintentionally create conditions that not only promote survival, but also the propagation, multiplication and spread of residual cancer cells. Such surgical-induced physiological changes are many and may include among which inflammation, tissue hypoxia, angiogenesis, surgical stress response, and immunosuppression [118]. All opioids, volatile and local anaesthetics are able to influence, directly or indirectly, the rate of proliferation, epithelial-mesenchymal transition, and the invasiveness properties of cancer cells as well as various elements of the tumour microenvironment. Nevertheless, none of these drugs has proven to have a direct causal relationship between their use in the perioperative period and a diminution of cancer recurrence rates or an increase in cancer-specific survival. As the last argument, stronger and more strict study protocols are needed to find consistent significant results.

In this point of view, a number of limitations can frequently be found in study designs and in recommendations that define how to solve these problems for future study design. The first limitation is matching for population and tumour confounders. It is important to note that these do not address the influence of the histopathological parameters such as tumour histology, size, margins, and the number of positive nodes as key factors that determines the risk of cancer recurrence. Another major drawback is that factors such as patient’s age, ethnicity, and gender, the patient’s general health status, and factors affecting the quality and efficiency of the surgical procedure such as the experience of the surgeons as well as patient pre, peri, and postoperative care have been scarcely investigated from the theoretical point of view. Apart from this is that the efficiency of metastatic detection techniques has been improved in recent years through the developing of novel methods for early detection, monitoring, and surveillance. Nonetheless, molecular-genetic imaging approaches allowing the visualisation and quantification of biochemical processes at the cellular and molecular level are needed to improve the tools and methodologies for monitoring circulating tumour cells. Additionally, when achieved, this might significantly enhance the power and accuracy of this approach for monitoring patient cancers noninvasively [119,120].

The second limitation is the transparency of drug dosing. In recent trials, there is transparency on the drugs used in the anaesthetic protocol, as well as documentation about perioperative consumption of possible relevant drugs. However, the dosing of applied drugs is left to the discretion of the treating anaesthesiologist. Since we now have various arguments on why the exact dose is also important. Since sevoflurane might have a different effect in low or high concentrations, knowing the dosage used perioperatively might help differentiate the effect on the outcome. Anaesthesia is rarely executed with a single anaesthetic, and study interventions might change dosing regiments of concomitant anaesthetics during surgery, further complicating interpretation of results. Additionally, dosing of analgesics compared to hemodynamic monitoring or nociceptive monitoring might give an idea of the relative surgical stress levels of the patient during surgery. As this may play an important role in cancer outcomes, collecting data on these surrogates might be interesting. Morphine consumption is frequently described as a measure for postoperative pain, but collected data perioperatively are generally lacking.

The third limitation is optimizing the study design for primary tumour or metastasis. A distinction must be made between targeting a primary tumour and targeting microsatellites or metastasis. As mentioned above, the microenvironment of the tumour is a complex synergistic interaction between many different cells. It is well known that concentrations and exposure are important factors in determining the effects of anaesthesia on cancer cells. Keeping the pharmacokinetics of anaesthetic drugs in mind, it may be very unlikely that a significant concentration of any kind is reached at the location of the primary tumour and its microenvironment. In vitro, it is obvious to reach the desired concentration of the drug at the target site, thus producing significant results. In the case of in vivo studies, this is more complicated and therefore troubled. Most studies that show anti-oncogenic effects on the primary tumour in vivo are animal studies dealing with very high nonclinical doses with a long exposure period [6,8,9,25,42,51,57,74,121]. Studies that apply clinically more comparable doses described more often an effect on metastasis and invasion than on reduction of the primary tumour [7,54,55,77,122]. It is possible that the in vivo concentration is higher intravenously or in highly vascularized areas. Additionally, circulating tumour satellites that are exposed to these concentrations are more strongly affected. In the clinical setting, the concentrations of anaesthetics are highest intravenously, depending on the diffusion coefficient of the drug used. This difference is mediated as diffusion to other tissues reaches equilibrium, but in practice, intravenous concentrations are higher. To conclude, it may be of major interest to harvest significant results if the focus is on the EMT and microsatellites leading to invasion and metastasis, since the concentrations of anaesthetics are highest in blood circulation and not at the site of the primary tumour.

The fourth limitation is the true importance of cancer tissue selection for research. In this literature research, several types of tissue were found that display different sensitivity to different anaesthetic drugs. For sevoflurane and lidocaine, different thresholds of a significant effect were found depending on the type of cell line and type of tissue [23,27,72,75,80]. This may play an important role in case selection in clinical studies. By limiting studies to only one type of tumour, and by correcting the tumour for known differences in differentiation or receptor expression, significant differences in outcome may be found when comparing anaesthetics.

## 5. Conclusions

There is evidence that anaesthesia can influence the biology of cancer through different hallmark mechanisms, although this is mostly described in vitro with the usage of very high concentrations that do not match daily clinical use. Most significant effects are only found in long durations of exposure, much longer than in most clinical anaesthetic procedures. For most anaesthetics, the effects are clearly correlated to drug concentration and time of exposure. These effects are likely dependent on various confounders, such as type of tumour, applied concentration, population, stress response and time of exposure. Yet, none of these have demonstrated a direct correlative causality between their use in the perioperative period and a reduction in cancer recurrence or an increase in cancer-specific survival. Moreover, a potential individual drug effect still remains complex to clarify in a clinical situation where different anaesthetics and drugs are similarly administered. In this review, no strict evidence of a significant clinical effect on cancer outcomes during surgery could be found. As a consequence, more transparent study protocols are absolutely necessary together with the exact description of drug dose and duration used during surgery for a better evaluation of the effect on cancer outcome, since these effects are strongly dose- and duration-dependent.

## Figures and Tables

**Table 1 cancers-15-00209-t001:** Studies describing the direct effects of midazolam, dexmedetomidine, and ketamine on cancer cells, the associated mechanisms of action and their respective pathways ^1^.

Midazolam				
Study	Type of Cancer	Effect on Cancer	Mechanism of Action	Pathway Described
Wang C. et al. [6]	Glioma, Lung	Anti-tumourigenic properties in very high doses. No effect in low (physiological) concentrations. Significant reduction in tumour size compared with tumours from naïve animals	Peripheral Benzodiazepine Receptor (PBR) on mitochondria resulting in reduction tumour burden, Ki67 expression and cyclin D expression	Intrinsic apoptotic pathway (exact mechanism unclear)
Qi Y. et al. [7]	Hepato-cellular	Inhibition of invasion and migration Repression of tumour growth	Overexpression of miR-124-3p and subsequent inhibition of PIM-1 resulting in cell cycle arrest and increased apoptosis	miR-124-3p/PIM- axis
Mishra SK. et al. [8]	Colon, leukemia	Growth inhibition of cancer cells Inhibited HT29 tumour growth in xenografts mice	Activation of caspase-9, capspase-3 and PARP indicating induction of the mitochondrial intrinsic pathway of apoptosis	Inhibition of pERK1/2 signalling leading to inhibition of the anti-apoptotic proteins Bcl-XL and XIAP and phosphorylation activation of the pro-apoptotic protein Bid
**Dexmedetomidine**				
**Study**	**Type of** **cancer**	**Effect on Cancer**	**Mechanism of Action**	**Pathway Described**
Wang C. et al. [6]	Glioma, Lung	Promotes cancer cell survival in vitro No significant effect in vivo	Increased Ki67 and cyclin D expression leading to cell proliferation	Via α2–adrenergic signalling and upregulation of antiapoptotic proteins Bcl-2 and Bcl-xL
Zhang P. et al. [9]	Esophagus	Suppressed tumour growth and metastasis	Increased apoptosis of esophageal cancer cells in vivo and in vitro	Upregulation of miR-143-3p and reducing the level of EPS8
**Ketamine**				
**Study**	**Type of** **Cancer**	**Effect on Cancer**	**Mechanism of Action**	**Pathway Described**
He H. et al. [10]	Breast	Decreased apoptosis	Upregulation of Bcl-2 expression	Non described

^1^ The correlated hallmark of cancer is given for reference on the role and importance in cancer development.

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
