# Peer review of "Does the Choice of Anaesthesia Affect Cancer? A Molecular Crosstalk between Theory and Practice"

_cancers, 2022, doi:10.3390/cancers15010209_

Round 1

Reviewer 1 Report

This manuscript summarizes the choice of anesthetics used for surgery for cancer resection, which has not been focused on much at the present. The authors quote various knowledge from publications and explain in detail the behavior of anesthetics on cancer cells, which are very helpful for readers. In particular, the published papers on each anesthetic are briefly summarized in the table, and they can be expected to support various research in the future.

Author Response

Dear Reviewer,

Thank you very much for your rewarded and useful comments. 

Reviewer 2 Report

The purpose of this review was to present an update of published data describing a potential influence of anesthesia on cancer outcome.

The authors focused first on direct effects of anesthetic drugs on cancer growth in vitro then tried to link these effects to  clinical survival observed after retrospective studies (the most numerous) second follow up or randomized trials.

The paper includes a huge amount of work and many references, presenting in vitro then in vivo effects for each drug class one after the other.

Unfortunately, the result is very long, hard to read and does not meet the requirements stated in the instructions to authors of « concise & precise update of the latest progress ».

Also, by presenting only direct effects and clinical outcome, it suggests to novice reader that there is no other than direct effects to influence the outcome, and therefore ignores all immune effects and inflammation influence of anesthetic agents. Finally they included proangiogenic effects which are not properly direct effects on cancer cells but works through circulating not only cellular mediators.

Two other issues may be discussed separately : one about the differences related to the type onf cancer, and the second about the dose-effect relationship of anesthetics on cancer outcome.

Issues about outcome, angiogenesis and clinical results have already been published including in this journal (Forget 2019, Enlund 2021). Therefore, I would suggest to the authors restricting the paper to the direct effects of anaesthetic drugs on cancer cells, which is an important issue enough to fill a nice paper.

In the discussion section, they can cite the above mentioned references and explain that the main difference between the lab and the OR is that, in the lab, one can study a single drug whereas in clinical practice no patient could be anesthetized with a single class of drugs.

Therefore a positive effect of one drug may hide the negative effect of the other. For example adding local anesthetics (intravenously or regionally) decreases opioid or hypnotic requirement and if a positive effect is observed, nobody can conclude if the benefit is due to the addition of local anesthetic or to the lack of others. As some of the authors are anesthesiologists, I am sure that they can comment on this issue in the discussion section.  

I would also suggest starting the review by a short general summary of all direct effects, proteins and mechanisms involved. Then tables for each drugs can be much more concise and much easier to read.

Author Response

Dear Reviewer,

Thank you very much for your rewarded review and useful comments concerning our manuscript. Our responses to these can be found hereunder. Furthermore, we adapted the manuscript in order to try to meet your comments as good as possible.

  1. "Also, by presenting only direct effects and clinical outcome, it suggests to novice reader that there is no other than direct effects to influence the outcome, and therefore ignores all immune effects and inflammation influence of anesthetic agents." Answer:  We initially limited our study to the direct effects on cancer cells, excluding other mechanisms like immunomodulation, inflammation and stress response. In line with your comment, we adjusted parts of the discussion to emphasize these limitations and to mention that not only direct effects play an important role in cancer outcome. 
  2. "Finally they included proangiogenic effects which are not properly direct effects on cancer cells but works through circulating not only cellular mediators." Answer: As for the proangiogenic effects which could work on circulating mediators, there is evidence that inflammatory and angiogenic pathways might directly influence cancer cells (for instance PMC3831877), so we included these in our results.
  3. "Issues about outcome, angiogenesis and clinical results have already been published including in this journal (Forget 2019, Enlund 2021). Therefore, I would suggest to the authors restricting the paper to the direct effects of anaesthetic drugs on cancer cells, which is an important issue enough to fill a nice paper." Answer: This very true. On the other hand, we tried to explain and illustrate that there are many conflicting data around in the literature leading to a lot of confusion and that no strong evidence of a significant clinical effect on cancer outcomes during surgery has been found. As a consequence of combining the direct effects and the issues as mentioned in your comment, we would hereby like to emphasize that more transparent study protocols are necessary together with the exact description of drug dose and duration used during surgery.
  4. "In the discussion section, they can cite the above mentioned references and explain that the main difference between the lab and the OR is that, in the lab, one can study a single drug whereas in clinical practice no patient could be anesthetized with a single class of drugs. Therefore a positive effect of one drug may hide the negative effect of the other. For example adding local anesthetics (intravenously or regionally) decreases opioid or hypnotic requirement and if a positive effect is observed, nobody can conclude if the benefit is due to the addition of local anesthetic or to the lack of others. As some of the authors are anesthesiologists, I am sure that they can comment on this issue in the discussion section.  " Answer: Thank you for this comment. We included the suggested references and we expanded on the subject of dosing of anesthetics, and included more specifically the remark on combining anaesthestics.
  5. "I would also suggest starting the review by a short general summary of all direct effects, proteins and mechanisms involved. Then tables for each drugs can be much more concise and much easier to read." Answer:  Thank you for this suggestion. In our vision, we have the feeling that it would be very difficult to provide a (short) list of proteins and pathways involved for interpretation of the results, since these vary depending on the type of cancer cells, and not all pathways are fully described in recent literature.
  6. "Two other issues may be discussed separately : one about the differences related to the type of cancer, and the second about the dose-effect relationship of anesthetics on cancer outcome." Answer: The differences related to the type of cancer are exstensively disscussed in the results per type of anaesthetic. The dose-effect relationship of anaesthetic is also reported in the results, but we have further expanded on the subject in the discussion, in particular on the dose-effect relationship of sevoflurane.
  7. "Unfortunately, the result is very long, hard to read and does not meet the requirements stated in the instructions to authors of « concise & precise update of the latest progress." Answer: We agree that the list of data is long and extensive, but the long list of references was necessary to fully cover the subject. We have tried to make the results as easily interpretable as possible, especially by sorting per anaesthetic, per type of tumor, and adding hallmarks of cancer for facilitating interpretation of mechanisms of action. 

Reviewer 3 Report

The manuscript of Debel et al entitled ‘Does the choice of anaesthesia affect cancer? A molecular cross-talk between theory and practice’ is a review of the putative impact of anesthetics on tumor outcomes based on basic science. This is a hot topic. However, major revisions are needed before acceptance.

The review suffers from a lack of a huge amount of fundamental research. Between 2010-2022, the number of basic science original articles dealing with anesthetics and cancer was: Dexmedetomidine (16), Midazolam (5), Volatiles (> 100), Propofol (> 100), Opioids (> 15), Local anesthetics (> 100)…

It is absolutely mandatory that the authors make an exhaustive review of the literature or focus their review on specific selected points. Some anesthetics are missing such as desflurane and some local anesthetics. Very recent publications, which have proven the immune antitumor effects of local anesthetics (including lidocaine) in vitro and in vivo need to be cited (for instance: PMID 35483744) because these publications involve major results. Which kind of opioids are described in this review: morphine? Sufentanil? Fentanyl? The ambivalence for sevoflurane (protumor effects or antitumor) has to be discussed. The authors described the epigenetic changes for propofol. However, many articles with fundamental research are also focused on the epigenetic changes mediated by volatiles, lidocaine and so on.

In sum, an important work with the literature has to be done before acceptance of the manuscript.

Minor points

‘between’ is written twice in the title

Please, make a choice between “anesthetics” and “anaesthetics” in the entire manuscript

Page 3 line 115: ‘administered’ is mistyping.

Page 20 line 297: ‘metastasize’ is mistyping

Author Response

Dear Reviewer, 

Thank you very much for your rewarded review and useful comments. Our answers to your comments can be found hereunder. We tried to adapt the manuscript as good as possible to your comments and suggestions. 

  1. "The review suffers from a lack of a huge amount of fundamental research. Between 2010-2022, the number of basic science original articles dealing with anesthetics and cancer was: Dexmedetomidine (16), Midazolam (5), Volatiles (> 100), Propofol (> 100), Opioids (> 15), Local anesthetics (> 100)… It is absolutely mandatory that the authors make an exhaustive review of the literature or focus their review on specific selected points." Answer:  Next to the huge amount of fundamental research, we are aware that the body of literature is vast, and the consulted studies in our manuscript are insufficient to describe all the possible effects that anaesthetics can have on cancer. Therefore, we limited our study to - and focused on - the direct effects on cancer cells, excluding other mechanisms like immunomodulation, inflammation and stress response. We adjusted the discussion to further emphasize these limitations. We only talk about indirect effects in our discussion of recent RCTs to explain that not only the direct effects have significant influence. "

  2. Some anesthetics are missing such as desflurane and some local anesthetics." Answer: On the topic of desflurane, our search query produced very little studies describing the direct effects of desflurane.  Our search method showed 2 studies of which the results were inconclusive. We have added the studies to our results. For local anaesthetics, we decided to focus on lidocaine because other anaesthetics like ropivacaine and bupivacaine are not applicable for intravenous use. Since we aim to study the direct effects, a high intravenous or local concentration of anaesthetic is required for maximal efficacy, whereas most other local anaesthetics are used in lower absolute doses compartmentalized around neuraxial regions or neural sheets, thus not producing the desired exposure for cancer cells.

  3. "Very recent publications, which have proven the immune antitumor effects of local anesthetics (including lidocaine) in vitro and in vivo need to be cited (for instance: PMID 35483744) because these publications involve major results." Answer: thank you for this comment. We have added the study by Bezu L et al. since it is indeed of major relevance. We have added a short discussion of local anaesthesia’s immune effects in discussing the RCT’s.
  4.  "Which kind of opioids are described in this review: morphine? Sufentanil? Fentanyl?" Answer: As for opioids, we have included the substances studied in the results. Most in vitro studies use morphine for µ-opiod receptor agonist. In clinical setting, morphine consumption is a frequently described parameter in study protocol, therefore ideal for our research. In most clinical studies dosage of sufentanyl or fentanyl is sparingly reported, making correct interpretation difficult. Furthermore, most patients receive a mix of opioids in perioperative period, complicating individual comparison.
  5.  "The ambivalence for sevoflurane (protumor effects or antitumor) has to be discussed." Answer: Thank you for this suggestion. We have further expanded on the subject in the disscusion section.
  6. "The authors described the epigenetic changes for propofol. However, many articles with fundamental research are also focused on the epigenetic changes mediated by volatiles, lidocaine and so on." Answer: Thank you for this comment. Indeed, we feel that the word 'epigenetics" was not the right term to describe the cellular effects in the context of the use of propofol as we are not describing the exact covalent posttranslational modifications such as acetylation, methylation or phosphorylation as part of a changed epigenetic code. Therefore, we corrected this in the manuscript into cellular effects. 

Reviewer 4 Report

Although the work is not a research article, it faithfully reflects some prerogatives of scientific rigor in evaluating the correlation between oncogenesis and anesthesia. the works analyzed to draft the revision have been extensively discussed with a critical spirit, deducing that both with volatile gases and other anesthetizers the probability of activating an ancogenesis is very unlikely. This was made possible only in in vitro experiments and with massive doses, much greater than the real intraoperative exposure.

Author Response

(The authors gave the same response as above.)

Round 2

Reviewer 2 Report

The aim of this revised review is to describe the mechanism of anesthetic drugs effects on cancer and cancer cells.

It is a huge amount of work and the message stating that dose and duration influence the results is important and too rarely emphasized.

Unfortunately compared to the first version, the authors modified very little the paper. Therefore, the main criticism I hade remain and the authors wrote a long response to explain that they agree with my suggestions but will not follow them.

The paper is still very long and difficult to read.

It is still missing an introduction describing the main mechanisms of toxicity against cancer cells because all readers may not be familiar with these details.

The paper describes direct cytotoxic effects, a little bit of angiogenesis and some mention of clinical outcome but not the immune effects, giving a truncated picture of the body response.

So I still recommend major revision of the paper, focusing on direct effects and dose response relationship, removing other issues.

Author Response

  • It is a huge amount of work and the message stating that dose and duration influence the results is important and too rarely emphasized. Dear reviewer, thank you for this suggestion, and we are more than happy that you support our vision on this complex matter. Therefore, we stressed this message stronger in the manuscript several times. We also changed wording to emphasize this message even more in a subtile way. 
  • It is still missing an introduction describing the main mechanisms of toxicity against cancer cells because all readers may not be familiar with these details. Dear reviewer,

    We have added a section in our introduction further expanding on the mechanisms of cancer development, hallmarks of cancer, and how we use them to describe the main mechanisms by which anaesthetics affect cancer cells.

  • The paper describes direct cytotoxic effects, a little bit of angiogenesis and some mention of clinical outcome but not the immune effects, giving a truncated picture of the body response

Dear reviewer, we are aware that the body of literature is vast, and the consulted studies in our manuscript are insufficient to describe all the possible effects that anaesthetics can have on cancer. We adjusted the discussion to further emphasize these limitations. We only talk about indirect effects in our discussion of recent RCTs to explain that not only the direct effects have significant influence. We have also expanded our description of the importance of immune function in cancer progression and how anaesthesia might influence this.

  • The paper is still very long and difficult to read. Dear  reviewer, we have tried to find a balance between all information available in the accessible literature and the length of the manuscript. Due to the 'hotness' of this topic, a lot of (very) different outcomes are available combined with even differing study set-ups. In this point of view we  honoustly feel that it is not that easy to produce a 'short' report. Although the overal end conclusion can be formulated shortly and concise, the basement to do this on soundly, needs a huge amount of work and inclusion of studies which is compromising the smaller amount of words. We need to be very complete in our argumentation. We truly hope this is not a stumbling stone to publish this huge amount of work we made. On the other hand, we performed a lot of editing in order to easify the reading of this (complex) work (combination of pathology and anesthesiology, which is somehow unique and maybe unexpected, but still is of much added value concerning this topic). We hope hat this manuscript wil trigger as well anesthesiologists as pathologists and other actors in cancer research in order to cooperate and explore more this topic / issue in the near future. 

Reviewer 3 Report

The authors have well addressed the comments of the reviewer.

Just add "desflurane" in the title of the table 2 ("direct effects of isoflurane, desflurane and sevoflurane...").

Author Response

Dear Reviewer,

Many thanks again for your much rewarded effort to review our revised manuscript. We made the changes as you suggested in your comments. 

Round 3

Reviewer 2 Report

The purpose of this paper was to put in balance the biological effects of anesthetic drugs on cancer development with their possible clinical influence on outcome. It is a huge amount of work with an extensive bibliographic review.

Unfortunately, the result is very long, boring and difficult to read, while several biological effects are still missing for some drugs. For example, very few immune effects are described for hypnotics whereas they are mentioned for opioids. Conversely angiogenesis is not detailed for opioids.

Second, the balance between biological and clinical effects examined for every drug is tricky because clinical protocols always combined several drugs (at least an hypnotic and an analgesic, sometimes several) and it is always difficult to distinguish a direct positive effect of a drug from its influence on other drugs sparing or from the proper effect of these other drugs.

Third, the mechanisms of toxicity should be further described to introduce the effects on each class of drugs. Some effort have been made to describe the mechanism of cell toxicity in the introduction. But it is not detailed enough and does not describe the factors that will be detailed drug by drug. For example HIF-1A appears in volatile toxicity. But what does HIF1A do? Same comment few lines later for SMAD3 signaling, CXCR2, P38, MAPK…. Where are they involved???

 I am aware how difficult it is to write such a review. But the result should be useful for the readers. That is why I suggested to the authors to focus on direct effects, difference related to the type of tumors and dose response relationship, and forget the clinical effects to make the paper shorter, strong enough and easy to read.

I obviously did not convince since very little has changed from the first, and the second version.

I still think that describing biological and clinical effects is too much for a single review and should be deeply modified

Author Response

Dear reviewer,

Many thanks again for reviewing our revised manuscript. We took notice of your comments / suggestions. Please find our answers hereunder.

Comment / Suggestion 1:

The purpose of this paper was to put in balance the biological effects of anesthetic drugs on cancer development with their possible clinical influence on outcome. It is a huge amount of work with an extensive bibliographic review.

Answer 1:

Thank you very much for appreciating this huge amount of work. We tried to reflect an up-to-date situation as much as possible connecting the 'hypothetical' in vitro situation with the daily routine clinical practice.

Comment / Suggestion 2:

Unfortunately, the result is very long, boring and difficult to read, while several biological effects are still missing for some drugs. For example, very few immune effects are described for hypnotics whereas they are mentioned for opioids. Conversely angiogenesis is not detailed for opioids.

Answer 2:

Thank you for this comment. We modified the manuscript and tables in order to make it better and easier to read. We understand that the summation of all scientific facts we found in the literature could make the text somewhat boring, but in our point of view the strength of this paper is the combination of  'basic science', pathological and anaesthetical view on this topic in general (not restricted to one cancer). We are aware that not all immune  and angiogenetic effects are described in all details as we limited our search to the most significant articles. Still we made changes and comments in the tables about these suggested topics where of importance.

Comment / Suggestion 3:

Second, the balance between biological and clinical effects examined for every drug is tricky because clinical protocols always combined several drugs (at least an hypnotic and an analgesic, sometimes several) and it is always difficult to distinguish a direct positive effect of a drug from its influence on other drugs sparing or from the proper effect of these other drugs.

Answer 3:

We are aware of this, and we clearly stated this - in other words - in the text.

Comment / Suggestion 4:

Third, the mechanisms of toxicity should be further described to introduce the effects on each class of drugs. Some effort have been made to describe the mechanism of cell toxicity in the introduction. But it is not detailed enough and does not describe the factors that will be detailed drug by drug. For example HIF-1A appears in volatile toxicity. But what does HIF1A do? Same comment few lines later for SMAD3 signaling, CXCR2, P38, MAPK…. Where are they involved???

Answer 4:

Thank you for this comment / suggestion. We are aware of the 'jungle' of proteins involved in different (oncogenic) pathways. We tried to create the context of these as much as possible in our tables, also for the novice reader.

Comment / suggestion 5:

I am aware how difficult it is to write such a review. But the result should be useful for the readers. That is why I suggested to the authors to focus on direct effects, difference related to the type of tumors and dose response relationship, and forget the clinical effects to make the paper shorter, strong enough and easy to read.

I obviously did not convince since very little has changed from the first, and the second version.

I still think that describing biological and clinical effects is too much for a single review and should be deeply modified

Answer 5:

Writing this type of manuscript is a huge amount of work. We tried to make the text much better readable, we adapted our tables and texts in this way. We still think that linking the biological and the described clinical effects is of added value to the reader (anaesthesiologist,  pathologist, molecular biologist, oncologist), especially as we did  not restrict to one or some types of cancer.  And the latter one is one of the key messages in our conclusion regarding variable results due to tumour type. Despite this,  we are aware that this manuscript is somehow technical, but still is very usable for the novice but also the molecular biological/oncological-trained reader, more specifically to motivate futural (basic and clinical) research.  
